# Design, Synthesis and Antiproliferative Evaluation of Bis-Indole Derivatives with a Phenyl Linker: Focus on Autophagy

**DOI:** 10.3390/molecules28010251

**Published:** 2022-12-28

**Authors:** Budovska Marianna, Michalkova Radka, Kello Martin, Vaskova Janka, Mojzis Jan

**Affiliations:** 1Department of Organic Chemistry, Institute of Chemistry, Faculty of Science, Pavol Jozef Šafárik University, 040 01 Košice, Slovakia; 2Department of Pharmacology, Faculty of Medicine, Pavol Jozef Šafárik University, 040 01 Košice, Slovakia; 3Department of Medical and Clinical Biochemistry, Pavol Jozef Šafárik University, 040 01 Košice, Slovakia

**Keywords:** indole phytoalexins, cell death, autophagy, GSH, lung cancer

## Abstract

This work deals with the study of the synthesis of new bis-indole analogues with a phenyl linker derived from indole phytoalexins. Synthesis of target bis-indole thiourea linked by a phenyl linker was achieved by the reaction of [1-(tert-butoxycarbonyl)indol-3-yl]methyl isothiocyanate with p-phenylenediamine. By replacing the sulfur of the thiocarbonyl group in bis-indole thiourea with oxygen using mesityl nitrile oxide, a bis-indole homodimer with a urea group was obtained. A cyclization protocol utilizing bis-indole thiourea and methyl bromoacetate was applied to synthesize a bis-indole homodimer with a thiazolidin-4-one moiety. Bis-indole homodimers derived from 1-methoxyspirobrassinol methyl ether were prepared by bromospirocyclization methodology. Among the synthesized analogues, compound 49 was selected for further study. To evaluate the mode of the mechanism of action, we used flow cytometry, Western blot, and spectroscopic analyses. Compound 49 significantly inhibited the proliferation of lung cancer cell line A549 with minimal effects on the non-cancer cells. We also demonstrated that compound 49 induced autophagy through the upregulation of Beclin-1, LC3A/B, Atg7 and AMPK and ULK1. Furthermore, chloroquine (CQ; an autophagy inhibitor) in combination with compound 49 decreased cell proliferation and induced G1 cell cycle arrest and apoptosis. Compound 49 also caused GSH depletion and significantly potentiated the antiproliferative effect of cis-platin.

## 1. Introduction

Cancer is one of the leading causes of death in both developing and developed countries [1]. The International Agency for Research on Cancer estimated almost 20 million new cancer cases and nearly 10 million cancer deaths worldwide in 2020. Among different cancer types, lung cancer (LC) was the leading cause of cancer death in 2020 in both sexes, as projected in GLOBOCAN 2020 [2].

Lung cancer therapy depends on the type of cancer. Regardless of the type of LC, chemotherapy is often involved in the treatment protocol. However, it has several limitations such as toxicity to non-cancer cells or chemoresistance induction [3]. Thus, it is critical to develop an effective and safe therapy to combat cancers including LC.

Phytochemicals are of great interest because they have been shown to be effective and inexpensive with minimal side effects [4]. Indole phytoalexins, cruciferous-derived phytochemicals, are secondary metabolites produced by plants after their exposure to biotic or abiotic stresses [5]. In addition to their plant-defensive functions, several indole phytoalexins, either natural or synthetic, have been evaluated for their growth inhibitory effect using cancer cell models [6]. We and others documented the growth-inhibitory effect of several natural and synthetic indole phytoalexins in various cancer lines such as colon, breast, cervix, liver, and prostate [7,8,9,10,11,12,13,14]. Bis-indole alkaloids from natural sources (e.g. 3,3′-diindolylmethane, hyrtiosin B) [15,16] and synthetic bis-indole hybrids [17,18] acted as very powerful inhibitors of growth for many types of human tumor cells in vitro. Bis-indole derivatives often showed enhanced antiproliferative and anticancer effects against a wide range of human cancer cells compared to their corresponding monomers [19].

Considering the above facts and in conjunction with our research work on indole phytoalexins as anticancer agents, we sought to design and synthesize bis-indoles inspired by indole phytoalexins that have in vitro antiproliferative activity. Our design was based on the linkage of two peripheral indole-based monomeric units with a phenyl linker to obtain the target bis-indole analogues of indole phytoalexins.

## 2. Results and Discussion

### 2.1. Chemistry

The synthetic pathway proposed to obtain the target bis-indoles 48 and 49 is depicted in Figure 1. In the synthesis of bis-indole thiourea 48 linked by a phenyl linker, we decided on a strategy based on the reaction of [1-(tert-butoxycarbonyl)indol-3-yl]methyl isothiocyanate (**47**) with p-phenylenediamine (**30**). [1-(tert-Butoxycarbonyl)indol-3-yl]methyl isothiocyanate (**47**) was prepared following a previously published procedure [20]. During a 24 h reflux of two equiv. of isothiocyanate **47** and one equiv. of p-phenylenediamine dihydrochloride (**30**) in dichloromethane in the presence of 6 equiv. triethylamine, we managed to prepare the corresponding bis-indole homodimer **48** in a yield of 69% (Figure 1). The structure of bis-indole thiourea 48 was confirmed by NMR spectroscopy (Appendix A). In the ^13^C NMR spectrum, a chemical shift at 180.8 ppm could be observed, proving the presence of a C=S group.

With the aim of finding an effective anti-cancer agent, we used the obtained bis-indole thiourea **48** as a starting substrate for the preparation of bis-indole urea **49**. The replacement of the sulfur of the thiourea moiety with oxygen was carried out using mesityl nitrile oxide in a mixture of absolute solvents chloroform/methanol (1:1) for 2 h at room temperature. The desired bis-indole urea **49** precipitated from the reaction mixture as white crystals in a yield of 58% (Figure 1). For the prepared urea **49**, a chemical shift for the C=O group of the urea skeleton at a value of 155.7 ppm was observed in the ^13^C NMR spectrum, confirming the conversion of thiourea to urea (Appendix A).

Bis-indole thiourea **48** was treated with methyl bromoacetate in the presence of triethylamine as a base. During the reaction of bis-indole thiourea **48** with methyl bromoacetate, the formation of two possible regioisomers **51a** and **51b** was considered. However, the cyclization reaction of homodimer **48** lasting 24 h gave only thiazolidin-4-one **51a** in 57% yield (Figure 2). The product was purified by column chromatography followed by crystallization from an acetone/*n*-hexane. The structure of the product was determined using 2D NMR spectra (Appendix A). In the HMBC spectrum, a correlation was observed between the CH2 group (bound to indole, δ 5.14 ppm) and the C=O group of the thiazolidin-4-one ring (δ 171.5 ppm), which proved that it was a derivative **51a**.

By bromospirocyclization of bis-indole thiourea **48**, we prepared a bis-indole homodimer **52** derived from the indole phytoalexin 1-methoxyspirobrasinol methyl ether, which served as a natural model. In the synthesis of this type of derivative, a cyclization protocol was chosen for the cyclization of thiourea **48** using bromine (1.1 equiv.) as the cyclization reagent, a mixture of absolute solvents dichloromethane/methanol (9:1), with methanol also acting as a nucleophile and triethylamine as a base. The reaction of the initially formed spiroindoleninium intermediate **B** with methanol led to the formation of the final diastereoisomers *trans*-(±)-**52a** and *cis*-(±)-**52b** in the ratio 73:27 (Figure 3). After separation of the diastereoisomers by column chromatography using eluent dichloromethane/ethyl acetate/ammonia 20:20:0.1, *trans*-diastereoisomer *trans*-(±)-**52a** was isolated as a pure substance in 55% yield, and *cis*-(±)-diastereoisomer *cis*-(±)-**52b** containing *trans*-diastereoisomer *trans*-(±)-**52a** as an inseparable impurity (20% yield). The determination of the *trans*- and *cis*-diastereoisomer ratio was carried out by integration of clearly resolved signals belonging to the H-2, Ha and Hb protons in the ^1^H NMR spectrum of the crude reaction mixture (Appendix A). Using the 2D NOESY experiment, the structure of the diastereoisomers *trans*-(±)-**52a**- and *cis*-(±)-**52b** was determined. The structure of the *trans*-diastereoisomer *trans*-(±)-**52a** was proved through the observed interaction between the Hb proton and the OCH_3_ protons. The detected interaction between the Hb proton and the H-2 proton in the NOESY spectrum was a confirmation of the *cis*-diastereoisomeric structure of the compound *cis*-(±)-**52b**.

### 2.2. Biological Activity

#### 2.2.1. MTT Screening Assay

The ability of newly synthesized indole phytoalexins to suppress cell proliferation was screened in vitro on eight cancer cell lines including human colorectal carcinoma (Caco-2 and HCT116), human cervical cancer (HeLa), human breast adenocarcinoma (MCF-7, MDA-MB-231), human alveolar adenocarcinoma (A549) and human leukemic T cell lymphoma (Jurkat), and two non-cancer cell lines (MCF-10A, Cos-7). Among the studied compounds, compounds **49** and **52a** showed the strongest antiproliferative activity against several cancer cell lines (Table 1). However, because compound **52a** also significantly decreased non-cancer cell proliferation, it was excluded from further studies. On the other hand, compound **49** showed greater selectivity against cancer cell lines, and it was selected for further analyses at a concentration of 20 μM, specifically on A549 cells.

#### 2.2.2. BrdU Cell Proliferation Assay

Measurement of the BrdU, a thymidine analogue, is a sensitive method used for the detection of DNA synthesis during the S phase of the cell cycle. Our results showed that compound **49** suppressed lung cancer cell proliferation in a dose-dependent manner (Figure 1) with an IC_50_ value of 16.1 μM after 72 h of incubation.

#### 2.2.3. Cell Cycle Analysis

To examine whether the antiproliferative effect of compound **49** is associated with cell cycle arrest, flow cytometry analyses of cell cycle progression were performed. As shown in Figure 2, indole phytoalexin **49** induced non-significant changes in the G1, S or G2/M phases of the cell cycle. Analyses of the sub-G0/G1 subpopulation (a marker of apoptosis) revealed minimal changes compared to DMSO control at all treatment time points.

#### 2.2.4. Annexin V/PI Staining

Phosphatidylserine (PS), normally localized in the inner leaflet of the cell membrane, is translocated on the outer surface during induction of apoptosis [21]. Annexin V/PI double staining was performed to analyze the externalization of PS on the outer plasmatic membrane surface together with cell population diversification (An^−^/PI^−^—living cells; An^+^/PI^-^ early apoptotic cells; An^+^/PI^+^—late apoptotic/necrotic cells, and An^−^/PI^+^—dead cells). As shown in Figure 3 indole phytoalexin **49** treatment did not induce total PS externalization compared to the DMSO-treated control.

#### 2.2.5. Effect of Compound **49** on the Expression of Autophagy-Related Proteins

On the basis of the above-mentioned results, we may say that compound **49** significantly inhibited A549 proliferation. However, surprising results of cell cycle analyses and Annexin V/PI staining showed that apoptosis was probably not involved in the suppression of cancer cell growth. Due to this, we decided to study the possible role of autophagy in indole phytoalexin-induced inhibition of cell proliferation. Autophagy is a pro-survival process that allows the recycling of unnecessary or damaged cellular components and plays an important role in cell homeostasis. Moreover, autophagy may also prevent apoptosis of cancer cells [22], and it is suggested that autophagy suppression may initiate apoptosis of cancer cells exposed to anticancer drugs [23] or experimental agents [24].

*Protein p62*, also known as sequestosome 1, is a multifunctional, classic autophagy receptor, considered together with LC3AB as an important marker of ongoing autophagic flux. It is involved in cell signaling, metabolism, oxidative stress and cell death. Its main function is the degradation of misfolded proteins through the ubiquitin–proteasome system (UPS) and autophagy, thereby maintaining intracellular proteostasis [25]. P62 binds the autophagosomal membrane protein LC3 and brings p62-containing cytoplasmic contents to the autophagosome for degradation. Our results showed a significant increase in p62 expression in compound **49**-treated lung cancer cells in all time periods (*** *p* < 0.001) (Figure 4 and Figure 5). This protein is normally degraded during autophagy after interaction with LC3 II [26]. On the other hand, accumulation of p62 has also been cited in cancer cells exposed to a variety of potential anticancer compounds [27,28,29].

*Beclin-1* plays a key role in the autophagy process, which is activated in cells during periods of stress. It ensures the recruitment of autophagosomal proteins to the pre-autophagosomal structure (PAS) and is part of a complex that is essential for the induction of autophagy (VSP34 complex). It also interacts with proteins from the Bcl-2 family, by which it can be inactivated or cleaved by caspases. This is considered a crosstalk between autophagy and apoptosis [30]. Western blot analyses showed that compound **49** significantly upregulated Beclin-1 in all time periods, most significantly after 24 h of incubation (*p* < 0.05, *p* < 0.01, *p* < 0.001) (Figure 4 and Figure 5). There is no mention in the literature focusing on Beclin-1 (and even autophagy) and indole phytoalexins. However, another phytoalexin, resveratrol, has been described to stimulate autophagy in oral cancer cells. Among several autophagy markers, Beclin-1 expression was upregulated in resveratrol-treated cells [31]. Later, Fan et al. [32] confirmed the ability of resveratrol to induce autophagy associated with upregulation of Beclin-1 also in lung cancer A549 cells.

*ULK1/AMPK.* ULK1 is a serine/threonine protein kinase. The ULK1 complex controls the formation of the phagophore, the initial autophagosomal precursor membrane structure. Upon induction of autophagy, the ULK1 complex translocates to autophagy initiation sites and regulates the recruitment of a second kinase complex, VPS34. ULK1 can be inhibited by phosphorylation by mammalian target of rapamycin complex 1 (mTORC1) or activated after AMP-activated protein kinase (AMPK) phosphorylation [33]. Under conditions of nutrient deficiency, AMPK is activated, which directly activates ULK1. The kinase is phosphorylated, i.e., activated by an increased AMP/ATP ratio due to cellular and environmental stress. AMPK also inactivates mTORC1 and thus blocks its inhibitory action on autophagy activation [34]. As shown in Figure 4 and Figure 5, the exposure of A549 cancer cells to **49** led to significant upregulation of the phosphorylated form of ULK1 at Ser555 in all time periods, mostly after 24 and 48 h of incubation (*p* < 0.05, *p* < 0.01). Furthermore, a significant increase in the phosphorylated form of AMPK was also observed (*p* < 0.01, *p* < 0.001). Similar results were also demonstrated by Ko et al. [35], who found that pterostilbene, a naturally occurring phytoalexin, induced autophagy in oral cancer cells, as confirmed by increased expression of ULK and AMPK. Because activated (i.e., phosphorylated) AMPK phosphorylates ULK1 at Ser555 during autophagy initiation [36], we suggest that compound **49** can initiate autophagy in A549 lung cancer cells.

*LC3A/B,* a microtubule-associated protein 1A/1B-light chain 3 (LC3), is a soluble intracellular protein. After synthesis of the pro-LC3 protein, this protein is first cleaved to form its cytosolic form known as LC3-I. The cytosolic form of LC3 (LC3-I) conjugates with phosphatidylethanolamine to form the LC3-phosphatidylethanolamine conjugate (LC3-II), which is recruited to autophagosomal membranes. After the formation of autophagosomes, their fusion with lysosomes occurs, autolysosomes are formed, and intra-autophagosomal material (cytoplasmic components including cytosolic proteins and organelles) is degraded by lysosomal hydrolases. Changes in the levels of the autophagosomal marker LC3-II are considered a sign of ongoing autophagy, and its lysosomal turnover thus reflects autophagic activity [37]. Results of our experiments (Figure 4 and Figure 5) showed that indole phytoalexin **49** significantly increased LC3A/B protein expression in A549 cells in all time periods (*p* < 0.01, *p* < 0.001). Association of LC3A/B up-regulation and autophagy induction has also been referred previously [38,39,40].

*Atg7* belongs to the family of Atg (autophagy-related) proteins involved in the process of autophagy. A ubiquitin-like conjugation system is required for autophagosome formation. The Atg7 protein is an E1-like activating enzyme that ensures the conjugation reaction and is essential for successful conversion of cytosolic LC3-I to its conjugated form, LC3-II. [41]. We observed (Figure 4 and Figure 5) that the levels of Atg7 were significantly increased in compound **49**-treated A549 cells (*p* < 0.05, *p* < 0.01, *p* < 0.001). Similar results were reported earlier. Hseu et al. [42] studied the effect of a natural chalcone, flavokawain B, on A549 lung cancer cells. Among other things, they found that flavokawain B induced autophagy associated with up-regulation of LC3-II and Atg7. Similarly, Maiti et al. [43] found that application of curcumin to cancer cells induced autophagy associated with a significant increase in Atg7 levels with concomitant increase in LC3A/B expression. Moreover, in curcumin-induced autophagy, up-regulation of beclin-1 has also been observed. Among phytoalexins, resveratrol has been found to induce autophagy in breast cancer stem-like cells, as detected by increased expression of autophagy markers including Beclin-1, LC3 II and Atg7 [44].

Furthermore, to confirm or exclude the role of autophagy in the antiproliferative effect of compound **49**, we performed a set of experiments with chloroquine, an autophagy inhibitor.

#### 2.2.6. MTT Assay with Chloroquine Combination

As shown in Figure 6, chloroquine (CQ) at a doses of 5, 10 and 20 μM significantly potentiated the effect of compound **49**, most effectively at the highest concentration used (*p* < 0.001 vs. DMSO, vs. **49** alone as well as vs. CQ alone). These results prompted us to study the effect of the CQ/compound **49** combination on cell cycle and apoptosis induction.

#### 2.2.7. Effect of Chloroquine Co-Treatment on Cell Cycle

When compared to compound **49** alone, a more significant effect in combination with CQ was observed after 24 h of incubation. As shown in Figure 7, compound **49** in combination with CQ induced significant cell cycle arrest at the G1 phase after 24 h of incubation with a concomitant decrease in cells in S phase when compared to DMSO, CQ or compound **49** alone. Analyses of the subG0/G1 subpopulation revealed non-significant changes compared to the DMSO control and to all other samples at all treatment time points.

#### 2.2.8. Effect of Chloroquine on Apoptosis

In comparison with compound **49** alone, combination with CQ significantly increased the number of cells in late apoptosis compared to DMSO, CQ or compound **49** alone at all treatment time points (Figure 8).

#### 2.2.9. Effect of Compound **49** on Glutathione (GSH) Levels and Activity of GSH-Related Enzymes

Glutathione is an important intracellular molecule involved in carcinogen detoxification. On the other hand, its elevated state in cancer cells is often associated with resistance to chemotherapy [45]. Furthermore, we previously found that some derivatives of natural indole phytoalexins possess remarkable glutathione-depleting potency [10,46]. Our results showed that treatment of A459 lung cancer cells with compound 49 significantly reduced levels of GSH with a concomitant increase in glutathione reductase (GR) and glutathione-S-transferase (GST) activity in all time periods. On the other hand, the activity of superoxide dismutase (SOD) showed no significant changes in compound **49**-treated A549 cells, and the activity of glutathione peroxidase (GPx) even significantly decreased after 24 and 48 h of incubation (Table 2). Because high levels of GSH are associated with cancer cell resistance, the GHS-depleting effect of compound 49 could be interesting due to new cancer chemosensitizer development. In the case of GST and GR, we hypothesize that an increase in the activity of both enzymes may be a compensatory mechanism induced by GSH depletion [10,46]. Furthermore, several lines of evidence showed an association between GHS depletion and autophagy initiation in both cancer and non-cancer cells [10,46,47,48]. On the basis of the above mentioned, we suggest that GSH depletion can be involved in compound **49** autophagy induction.

#### 2.2.10. Effect of Simultaneous Treatment with Compound **49** and Cis-Platin (CisPt) on A549 Human Lung Cancer Cell Proliferation

Furthermore, the depletion of GSH has been documented to potentiate the effect of anticancer drugs [49,50]. Glutathione plays an important role in cisPt detoxification, so depletion of GSH is believed to potentiate the cisPt-induced DNA damage and increase the chemosensitivity of cancer cells to this anticancer drug [51]. Due to the GSH-depleting effect of compound **49**, we performed a set of experiments with cis-platin (CisPt) alone or in combination with compound **49** (Figure 9). Treatment of lung cancer cells with different CisPt concentrations alone for 72 h resulted in dose-dependent growth inhibition with the maximal effect of CisPt at the concentration of 10.0 μM (39% reduction of cell growth). Simultaneous treatments with CisPt (10 μM) and 20 μM of compound **49** caused a 59% reduction in cell growth. Statistical analysis showed that the drug combination was more effective than compound **49** or CisPt treatment alone (*p* < 0.05; *p* < 0.01). These data suggest that compound **49** potentiates the cancer cell suppressive effect of cisPT, and this effect can be due to the depletion of cellular GSH. Similar results we also obtained by Kachadourian et al. [52], who found that selected flavonoids potentiated cytotoxicity of cis-platin in A549 lung cancer cells via depletion of GSH.

In summary, in this study, a new series of bis-indole analogues with a phenyl linker derived from indole phytoalexins were designed and synthesized. The structure of the analogues **48–52a** was confirmed by NMR spectroscopy. Compound **49** significantly suppressed the growth of lung cancer cells with low cytotoxicity in non-cancer cells. Surprisingly, the mechanistic analyses showed that compound **49** did not affect either the cell cycle or apoptosis. Subsequent analyses, however, showed that compound **49** initiated autophagy, as demonstrated by increased phosphorylation or expression of several autophagy markers, including AMPK, ULK1, Beclin-1, LC3A/B, p62 and Atg7. Furthermore, experiments with chloroquine (an autophagy inhibitor) showed cell cycle arrest at the G1 phase and increased numbers of cells in late apoptosis in combination with compound **49**. On the basis of these results, we suggest that autophagy probably has a defensive function, and it is not involved in the antiproliferative effect of compound **49**. In addition, compound **49** caused GSH depletion in cancer cells, resulting in increased sensitivity to cancer treatment, as documented by the potentiation of cis-platin cytotoxicity in indole phytoalexin-treated cells. These results provide new and interesting information that compound **49**-induced GSH depletion could be useful to increase the effectiveness of anticancer drugs.

## 3. Materials and Methods

### 3.1. General Remarks

^1^H NMR (400 MHz) and ^13^C NMR (100 MHz) spectra were measured in CDCl_3_ or DMSO-d_6_ on a Varian Mercury Plus spectrometer. Chemical shifts (δ) are reported in ppm downfield from TMS as internal standard, and coupling constants (*J*) are given in hertz (Hz). Melting points were determined on a Kofler micro melting point apparatus and are uncorrected. Infrared spectra were recorded on an Avatar FT-IR 6700 (Thermo Scientific, UK) using an attenuated total reflectance (ATR) method in the range 4000–400 cm^−1^. Microanalyses were performed with a Perkin-Elmer, CHN 2400 elemental analyzer. The progress of chemical reactions was monitored on TLC-sheets ALUGRAM^®^ SIL G/UV_254_ (Macherey-Nagel, Dueren, Germany). Detection was carried out with ultraviolet light (254 nm). Column chromatography was performed on silica gel, Kieselgel 60 Merck Type 9385 (0.040–0.063). All commercial reagents were used in the highest available purity without further purification.

### 3.2. Synthesis

#### 3.2.1. Synthesis of N,N′-(1,4-phenylene)bis{N′-[1-(tert-butoxycarbonyl)indol-3-yl]methyl (thiourea)} (**48**)

To a solution of {[1-(*tert*-butoxycarbonyl)indol-3-yl]methyl} isothiocyanate (**47**; 0.623 g, 2.16 mmol) and 1,4-phenylenediamine dihydrochloride (0.196 g, 1.08 mmol) in dichloromethane (10 mL), triethylamime was added (0.655 g, 0.903 mL, 6.48 mmol). The reaction mixture was refluxed for 24 h. The resulting white precipitate was filtered out, and the pure product was isolated. 

Yield: 69% (0.511 g), white crystals, m.p. = 204–206 °C (dichloromethane), *R_f_* (*n*-hexane/ethyl acetate 1:1) = 0.38.

^1.^ H NMR (400 MHz, DMSO-d_6_) δ: 9.53 (s, 2H, NH, NH′), 8.04 (d, *J* = 8.2 Hz, 2H, H-7, H-7′), 8.03 (s, 2H, NH, NH′), 7.75 (d, *J* = 7.8 Hz, 2H, H-4, H-4′), 7.62 (s, 2H, H-2, H-2′), 7.33 (dd, *J* = 8.2 Hz, *J* = 0.8 Hz, 2H, H-6, H-6′), 7.32 (s, 4H, H-2″, H-3″, H-5″, H-6″), 7.25 (ddd, *J* = 7.8 Hz, *J* = 7.8 Hz, *J* = 0.8 Hz, 2H, H-5, H-5′), 4.84 (d, *J* = 4.4 Hz, 4H, CH_2_, CH_2_′), 1.61 [s, 18H, (CH_3_)_3_, (CH_3_)_3_′].

^13.^ C NMR (100 MHz, DMSO-d_6_) δ: 180.8 (C=S, C=S′), 149.5 (C=O, C=O′), 135.9 (C-1″, C-4″), 135.3 (C-7a, C-7a′), 129.6 (C-3a, C-3a′), 125.0 (C-6, C-6′), 124.5 (C-2, C-2′), 123.9 (C-2″, C-3″, C-5″, C-6″), 123.1 (C-5, C-5′), 120.2 (C-4, C-4′), 118.4 (C-3, C-3′), 115.2 (C-7, C-7′), 84.3 [C(CH_3_)_3_, C(CH_3_)_3_′], 39.3 (CH_2_, CH_2_′), 28.1 [C(CH_3_)_3_, C(CH_3_)_3_′].

IR **ν**_max_ (cm^−1^) 3275 (NH), 2976, 1731 (C=O), 1508, 1451, 1368, 1255, 1152, 1086.

Anal. Calcd for C_36_H_40_N_6_O_4_S_2_ requires: C 63.13%; H 5.89%; N 12.27%; Found: 63.33%; H 6.06%; N 12.51%.

#### 3.2.2. Synthesis of N,N′-(1,4-phenylene)bis{N′-[1-(tert-butoxycarbonyl)indol-3-yl]methyl (urea)} (49)

To a solution of thiourea (**48**; 0.103 g, 0.15 mmol) in anhydrous chloroform/methanol (1.3 mL/1.3 mL), mesityl nitrile oxide (0.097 g, 0.6 mmol) was added. The reaction mixture was stirred for 2 h at room temperature. The resulting white precipitate was filtered out, and the pure product was isolated.

Yield: 58% (0.057 g), white crystals, m.p. = >300 °C (chloroform/methanol), *R_f_* (*n*-hexane/ethyl acetate 1:1) = 0.13.

^1.^ H NMR (400 MHz, DMSO-d_6_) δ: 8.29 (s, 2H, NH, NH′), 8.03 (d, *J* = 8.2 Hz, 2H, H-7, H-7′), 7.68 (d, *J* = 7.4 Hz, 2H, H-4, H-4′), 7.56 (s, 2H, H-2, H-2′), 7.34–7.30 (m, 2H, H-6, H-6′), 7.23 (s, 4H, H-2″, H-3″, H-5″, H-6″), 7.26–7.22 (m, 2H, H-5, H-5′), 6.42 (t, *J* = 5.4 Hz, 2H, NH, NH′), 4.39 (d, *J* = 5.4 Hz, 4H, CH_2_, CH_2_′), 1.60 [s, 18H, (CH_3_)_3_, (CH_3_)_3_′].

^13.^ C NMR (100 MHz, DMSO-d_6_) δ: 155.7 (C=O, C=O′), 149.5 (C=O, C=O′), 135.4 (C-7a, C-7a′), 134.7 (C-1″, C-4″), 129.6 (C-3a, C-3a′), 124.9 (C-6, C-6′), 123.8 (C-2, C-2′), 123.0 (C-5, C-5′), 120.1 (C-3, C-3′), 120.0 (C-4, C-4′), 118.9 (C-2″, C-3″, C-5″, C-6″), 115.2 (C-7, C-7′), 84.1 [C(CH_3_)_3_, C(CH_3_)_3_′], 34.6 (CH_2_, CH_2_′), 28.1 [C(CH_3_)_3_, C(CH_3_)_3_′].

IR **ν**_max_ (cm^−1^) 3306 (NH), 2976, 1735 (C=O), 1628, 1566 (NHC=O), 1369, 1354, 1258, 1157, 1085.

Anal. Calcd for C_36_H_40_N_6_O_6_ requires: C 66.24%; H 6.18%; N 12.88%; Found: 66.49%; H 6.37%; N 13.04%.

#### 3.2.3. Synthesis of 2,2′-[1,4-phenylenebis(azanylylidene)]bis{3-[(1-(tert-butoxycarbonyl)-indol-3-yl)methyl]-1,3-thiazolidin-4-one} (51a)

To a solution of thiourea (**48**; 0.103 g, 0.15 mmol) in anhydrous dichloromethane (4.5 mL), methyl bromoacetate (0.229 g, 0.142 mL, 1.5 mmol) was added. After 5 min, triethylamine (0.303 g, 0.418 mL, 3 mmol) was added, and the reaction mixture was stirred for 24 h at room temperature. After diluting the mixture with dichloromethane (15 mL), the dichloromethane layer was washed with saturated NaCl solution (2 × 15 mL). The organic layer was dried over anhydrous sodium sulphate. After evaporation of the solvent, the residue was purified by column chromatography on a silica gel (15 g SiO_2_, eluent *n*-hexane/ethyl acetate 2:1). The obtained product was crystallized from acetone/*n*-hexane.

Yield: 57% (0.065 g), white crystals, m.p. = 196–198 °C (acetone/*n*-hexane), *R_f_* (*n*-hexane/ethyl acetate 2:1) = 0.73.

^1.^ H NMR (400 MHz, CDCl_3_) δ: 8.14 (d, *J* = 7.5 Hz, 2H, H-7, H-7′), 7.95 (d, *J* = 7.5 Hz, 2H, H-4, H-4′), 7.84 (s, 2H, H-2, H-2′), 7.35–7.26 (m, 4H, H-5, H-5′, H-6, H-6′), 7.00 (s, 4H, H-2″, H-3″, H-5″, H-6″), 5.14 (s, 4H, CH_2_, CH_2_′), 3.79 [s, 4H, CH_2_ (thiazolidin-4-one), CH_2_ (thiazolidin-4-one)′], 1.67 [s, 18H, (CH_3_)_3_, (CH_3_)_3_′].

^13.^ C NMR (100 MHz, CDCl_3_) δ: 171.5 (C=O, C=O′), 153.9 (C=N, C=N′), 149.6 (C=O, C=O′), 144.6 (C-1″, C-4″), 135.2 (C-7a, C-7a′), 129.5 (C-3a, C-3a′), 127.2 (C-2, C-2′), 124.5 (C-6, C-6′), 122.7 (C-5, C-5′), 121.9 (C-2″, C-3″, C-5″, C-6″), 120.1 (C-4, C-4′), 115.1 (C-7, C-7′), 114.9 (C-3, C-3′), 83.8 [C(CH_3_)_3_, C(CH_3_)_3_′], 37.3 (CH_2_, CH_2_′), 32.7 [CH_2_ (thiazolidin-4-one), CH_2_ (thiazolidin-4-one)′], 28.2 [C(CH_3_)_3_, C(CH_3_)_3_′].

IR **ν**_max_ (cm^−1^) 3007, 1731 (C=O), 1717 (C=O), 1608, 1449, 1367, 1255, 1165.

Anal. Calcd for C_40_H_40_N_6_O_6_S_2_ requires: C 62.81%; H 5.27%; N 10.99%; Found: 62.97%; H 5.53%; N 11.19%.

#### 3.2.4. Synthesis of trans-(±)- and cis-(±)-N,N′-bis[1-(tert-Butoxycarbonyl)-2-methoxy-spiro{indoline-3,5′-[4′,5′]dihydrothiazol-2′-yl}]benzene-1,4-diamine [(±)-**52a**, (±)-**52b**]

A freshly prepared solution of bromine (0.855 mL, 0.365 mmol; the stock solution prepared by dissolving of bromine (0.04 mL) in anhydrous dichloromethane (1.76 mL)) was added to a solution of thiourea (48; 0.114 g, 0.166 mmol) in a mixture of anhydrous solvents dichloromethane/methanol (3.0 mL/0.3 mL). After stirring for 10 min., triethylamine (0.074 g, 0.102 mL, 0.730 mmol) was added, and the reaction mixture was stirred for another 5 min. After diluting the mixture with dichloromethane (20 mL), the dichloromethane layer was washed with saturated NaCl solution (2 × 25 mL). The organic layer was dried over anhydrous sodium sulphate. After evaporation of the solvent, the residue was submitted to chromatography on a silica gel (15 g, eluent dichloromethane/ethyl acetate/ammonia 20:20:0.1) to afford pure diastereoisomer *trans*-(±)-**52a** and *cis*-(±)-**52b** containing *trans*-(±)-**52a** as an inseparable impurity. The obtained *trans*-diastereoisomer *trans*-(±)-**52a** was crystallized from dichloromethane/*n*-hexane.

##### *Trans*-(±)-**52a**

Yield: 55% (0.068 g), white crystals, m.p. = 208–210 °C (dichloromethane/*n*-hexane), *R_f_* (dichloromethane/ethyl acetate/ammonia 20:20:0.1) = 0.50.

^1.^ H NMR (400 MHz, CDCl_3_) δ: 7.75 (s, 2H, H-7, H-7′), 7.37 (d, *J* = 7.5 Hz, 2H, H-4, H-4′), 7.25 (dd, *J* = 7.5 Hz, *J* = 7.5 Hz, 2H, H-6, H-6′), 7.03–7.00 (m, 6H, H-5, H-5′, H-2″, H-3″, H-5″, H-6″), 5.54 (s, 2H, H-2, H-2′), 4.34 (d, *J* = 12.1 Hz, 2H, H_b_, H_b_′), 4.07 (d, *J* = 12.1 Hz, 2H, H_a_, H_a_′), 3.54 (s, 6H, OCH_3_, OCH_3_′), 1.57 [s, 18H, (CH_3_)_3_, (CH_3_)_3_′].

^13.^ C NMR (100 MHz, CDCl_3_) δ: 159.2 (C=N, C=N′), 151.9 (C=O, C=O′), 142.2 (C-1″, C-4″), 141.5 (C-7a, C-7a′), 129.8 (C-6, C-6′, C-3a, C-3a′), 123.6 (C-4, C-4′, C-2″, C-3″, C-5″, C-6″), 121.6 (C-5, C-5′), 115.9 (C-7, C-7′), 98.6 (C-2, C-2′), 82.2 [C(CH_3_)_3_, C(CH_3_)_3_′], 65.4 (C-3, C-3′), 58.2 (OCH_3_, OCH_3_′), 53.8 (CH_2_, CH_2_′), 28.3 [C(CH_3_)_3_, C(CH_3_)_3_′].

IR **ν**_max_ (cm^−1^) 3325 (NH), 2889, 1717 (C=O), 1635, 1479, 1384, 1169, 1080.

Anal. Calcd for C_38_H_44_N_6_O_6_S_2_ requires: C 61.27%; H 5.95%; N 11.28%; Found: 61.50%; H 6.12%; N 11.54%.

##### *Cis*-(±)-**52b**

Yield: 20% (0.025 g), colorless oil, *R_f_* (dichloromethane/ethyl acetate/ammonia 20:20:0.1) = 0.59.

^1.^ H NMR (400 MHz, CDCl_3_) δ: 7.78 (s, 2H, H-7, H-7′), 7.41–7.38 (m, 2H, H-4, H-4′), 7.28–7.24 (m, 2H, H-6, H-6′), 7.06–7.01 (m, 6H, H-5, H-5′, H-2″, H-3″, H-5″, H-6″), 5.26 (s, 2H, H-2, H-2′), 3.92 (d, *J* = 12.1 Hz, 2H, H_b_, H_b_′), 3.63 (d, *J* = 12.1 Hz, 2H, H_a_, H_a_′), 3.56 (s, 6H, OCH_3_, OCH_3_′), 1.58 [s, 18H, (CH_3_)_3_, (CH_3_)_3_′].

### 3.3. Biological Activity

#### 3.3.1. Tested Compounds

All tested compounds were synthesized by Marianna Budovska (see above). The compounds were dissolved in DMSO. The final concentration of DMSO in the culture medium did not exceed 0.2% and had no cytotoxic effect.

#### 3.3.2. Cell Culture

The cell lines used for this study were Caco-2 (human colorectal adenocarcinoma cell line, American Type Culture Collection, Manassas, VA, USA (ATCC)), HCT116 (human colorectal carcinoma cells European Collection of Authenticated Cell Cultures, Salisbury, UK (ECACC)), HeLa (human cervical tumor cells, ECACC), MCF-7 (human breast tumor cells, ECACC), MDA-MB-231 (human breast tumor cells, ATCC), A549 (human lung carcinoma cells, ATCC), BLM (human melanoma lung metastases, a gift from prof. K. Smetana, Institute of Anatomy, Charles University in Prague), Jurkat (human leukemia cell line, ECACC), Cos-7 (immortalized monkey kidney fibroblasts, ATCC) and MCF-10A (human mammary epithelial cells, ATCC). Caco-2, MCF-7, BLM, Cos-7 and A549 cells were cultured in Dulbecco’s Modified Eagle’s Medium (DMEM) with high glucose and sodium pyruvate (GE Healthcare, Piscataway, NJ, USA); HCT116 lines, HeLa, MDA-MB-231 and Jurkat were cultured in RPMI-1640 medium; and MCF10A cells were cultured in high glucose DMEM/F12 (Dulbecco’s Modified Eagle’s Medium F12). The media were enriched with 10% fetal bovine serum (FBS) and a 1% solution of a mixture of antibiotics and antimycotics 1× HyClone™ (GE Healthcare, Chicago, IL, USA). DMEM/F12 medium was supplemented with epidermal growth factor (EGF) (20 ng/mL final), hydrocortisone (0.5 μg/mL final) and insulin (10 μg/mL final) (Sigma-Aldrich Chemie, Steinheim, Germany). The cells were placed in an incubator at 37 °C with humidified air containing 5% CO_2_.

#### 3.3.3. MTT Colorimetric Assay

The MTT assay is a colorimetric method used to quantify the metabolic activity of cells. MTT (3-(4,5-di-methylthiazol-2-yl)2,5-diphenyltetrazolium bromide) (Sigma-Aldrich Chemie, Steinheim, Germany) is reduced by oxidoreductases in the cytosol of cells to light-absorbing formazan, the presence of which can be measured spectrophotometrically and determines the half-maximal inhibitory concentration (IC_50_). A549 cells were seeded in 95-well culture plates at a density of 5 × 10^3^ per well. After 24 h of cultivation, the tested substance **49** was added in concentrations of 10, 50 and 100 µM.

We also incubated the cells with chloroquine at concentrations of 5–20 µM, which was added 1 h before the addition of the test substance, and cisplatin at concentrations of 3–10 µM. Cisplatin was added 24 h after the addition of 49. After 72 h of incubation, 10 µL of MTT (5 mg/mL) was added to each well. After 4 h, formazan crystals formed in the wells, which were dissolved in 100 uL of 10% SDS (sodium dodecyl sulfate). After 24 h, the absorbance was measured at a wavelength of 540 nm using an automated Cytation™ 3 Cell Imaging Multi-Mode Reader (Biotek, Winooski, VT, USA). These experiments were performed in three independent replicates.

#### 3.3.4. BrdU Cell Proliferation Assay

A549 tumor cells were plated at a density of 5 × 10^3^ cells/well in a volume of 80 µL (96-well plates). After 24 h of incubation in the respective medium, we added the indole phytoalexin **49** in concentrations of 5–30 µM. After 48 h of incubation with the test substance, we added BrdU labeling solution and incubated for another 24 h. Subsequently, the cells were fixed for 30 min with a fixing solution, aspirated and incubated with peroxidase-conjugated anti-BrdU antibody (100 µL, 1:100). After washing with washing buffer 3 times, TMB substrate solution (tetramethylbenzidine, Roche, Basel, Switzerland) was added in a volume of 100 uL to each well. During the reaction, 25 µL of 1M H_2_SO_4_ was added as a stop solution, and the amount of BrdU incorporated was detected with an automated Cytation™ 3 Cell Imaging Multi-Mode Reader (Biotek, Winooski, VT, USA) at a wavelength of 450 nm. Three independent analyses were performed.

#### 3.3.5. Cell Cycle Analysis

The A549 lung human alveolar adenocarcinoma cells were seeded at a density of 2 × 10^5^/dish. After 24 h incubation, the cells were treated with indole phytoalexin **49** at 20 µM concentration alone or after 1 h pre-treatment with chloroquine (20 µM), and the control group with DMSO (*v*/*v* 0.02%). The whole population of treated cells (adherent and floating) for cell cycle analyses was harvested at three different times (24, 48 and 72 h), washed in cold washing buffer (PBS), fixed in cold 70% ethanol and stored at −20 °C at least overnight. For flow cytometry analyses, cell suspensions were washed with PBS, resuspended in staining solution (Triton X-100 final concentration 0.1%, ribonuclease A final concentration 0.5 mg/mL, and propidium iodide final concentration 0.025 mg/mL) and incubated for 30 min in the dark at RT. Stained cells were analyzed using a BD FACSCalibur^TM^ Flow Cytometer (Becton Dickinson, San Jose, CA, USA) in FL-2 channel. Three independent analyses were performed.

#### 3.3.6. Annexin V/PI Analysis

The A549 lung human alveolar adenocarcinoma cells were seeded at a density of 2 × 10^5^/dish. After 24 h incubation, the cells were treated with indole phytoalexin **49** at 20 µM concentration alone or after 1 h pre-treatment with chloroquine (20 µM), and the control group with DMSO (*v*/*v* 0.02%). The whole population of treated cells (adherent and floating) was harvested at three different times (24, 48 and 72 h), washed with PBS and stained with Annexin V-Alexa Fluor^®^ 647 antibody (Thermo Scientific, Rockford, IL, USA) in binding buffer for 20 min in the dark at RT. After the wash step, samples were stained with 1 μL of propidium iodide (final concentration 25 μg/mL) for 5 min and analyzed using a BD FACSCalibur^TM^ Flow Cytometer (Becton Dickinson, San Jose, CA, USA). Three independent analyses were performed.

#### 3.3.7. Western Blot Analysis

A549 cells were seeded on Petri dishes (55 cm^2^) at a density of 1 × 10^6^ and allowed to culture in the medium. After 24 h, the test substance was added, and the cells were incubated for 24, 48 and 72 h. Subsequently, the cells were lysed with ice-cold Laemli buffer solution containing glycerol, 1M Tris/HCl (pH 6.8), 20% sodium dodecyl sulphate (SDS), deionized water and mixtures of phosphatase and protease inhibitors. After lysis, the cells were sonicated, and the protein concentration in the samples was quantified using the Pierce^®^ BCA Protein Assay Kit (Thermo Scientific, Rockford, IL, USA) and measured by an automated Cytation™ 3 Cell Imaging Multi-Mode Reader (Biotek) at a wavelength of 570 nm. Proteins, added in an amount of 30 µg per well were separated on 10% SDS-PAA gels for approximately 3 h (at a voltage of 100 V) and transferred to polyvinylidene difluoride) (PVDF) membranes using the iBlot™ 2 dry blotting system (Invitrogen, Carlsbad, CA, USA). The membranes were washed and blocked in a solution of 5% non-fat dry milk (Cell Signaling Technology^®^, Danvers, MA, USA) or 5% bovine serum albumin (BSA, SERVA, Heidelberg, Germany) in a solution of TBS-Tween (TBST, pH 7,4) at RT. After blocking non-specific binding, the membranes were incubated with primary antibody solutions overnight at 4 °C. The list of primary and secondary antibodies is given in Table 3. The next day, we washed the membranes 3 × 5 min in TBST solution and incubated with the appropriate secondary antibody conjugated with horseradish peroxidase (HRP) for 1 h at RT. We washed the membranes again with TBST (3 × 5 min). An iBright™ FL1500 Imaging System (Invitrogen, Carlsbad, CA, USA) using an ECL chemiluminescent substrate (Thermo Scientific, Rockford, IL, USA) was used to detect the expression of selected proteins. Densitometric analysis was performed using Image Studio™ Lite Software (LI-COR Biosciences, Lincoln, NE, USA). Beta-actin was used as a loading control, and each analysis was performed in three independent experiments.

#### 3.3.8. Measurement of Glutathione Content and Antioxidant Enzyme Activity

Both floating and adherent A549 cells were harvested at 24, 48, and 72 h after treatment with compound **49** (20 µM). SOD determination kit, Glutathione Peroxidase Cellular Activity Assay kit, Glutathione Reductase Assay Kit and Glutathione-S-transferase (GST) Assay Kit (all Sigma Aldrich, Germany) were used for the detection of relevant enzyme activity. Reduced glutathione (GSH) content was measured by the method originally described by Floreani et al. [53]. Assays were performed on an M 501 single beam UV/VIS spectrophotometer (Spectronic Camspec Ltd., Leeds, United Kingdom). All measured parameters were calculated per milligram or gram of protein determined using the bicinchoninic acid assay.

#### 3.3.9. Statistical Analysis

Results are expressed as mean ± standard deviation (SD). Statistical analyses of data were performed using standard procedures with one-way analysis of variance (ANOVA) followed by Bonferroni’s multiple comparison test. Differences were considered significant when *p* < 0.05.

## Data Availability

Not applicable.

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
