# Peer review of "Design, Synthesis and Antiproliferative Evaluation of Bis-Indole Derivatives with a Phenyl Linker: Focus on Autophagy"

_molecules, 2022, doi:10.3390/molecules28010251_

Round 1

Reviewer 1 Report

Comments

The article entitled “Design, synthesis and biological activity of bis-indole derivatives with a phenyl linker”. In this article the synthesis of new bis-indole analogues with a phenyl linker derived from indole phytoalexins was achieved. The sulfur group replacement of the thiocarbonyl group in bis-indole thiourea was achieved utilizing oxygen using mesityl nitrile oxide. Further cyclization between bis-indole thiourea and methyl bromoacetate was carried out for the synthesis of a bis-indole homodimer with a thiazolidin-4-one moiety. The bromospirocyclization process was utilized for the synthesis of bis-indole homodimers derived from 1-methoxyspirobrassinol methyl ether. The compound 49 was selected for further study for the inhibition of lung cancer cell line.

However, there are certain areas in this manuscript in which improvement is must needed.

1. Page 2, line 64: Please cite the article for the synthesis of 47.

2. Page 3, line 91: In HMBC spectrum the correlation between CH2 group bound to indole and C=N was observed but that doesn't prove the possibility of structure 51a because such correlation is also possible in 51b. However, the correlation between CH2 group and C=O group proves the formation of 51a. So, it is advised for the authors to correct the statement in terms of correlation between CH2 group and C=N.

Decision: Acceptance after minor revisions.

Author Response

We would like to thank the reviewer for his/her comments, which have given us the opportunity to improve the manuscript. Please sea attachment.

Reviewer 2 Report

The Author has done the Design, synthesis, and biological activity of bis-indole derivatives with a phenyl linker. The author synthesized 4 compounds and evaluated them against 10 cell lines. Compound 49 shows potent activity against A549 cells. Biological studies such as cell proliferation, cell cycle analysis and apoptosis assays were done to show its anti-cancer effect. The author has concluded from the cell cycle and annexin V assay that compound 49 doesn’t show an anticancer effect because of apoptosis. Western blot analysis of autophagy-related proteins suggests compound 49 initiates autophagy. To exclude the role of autophagy in the antiproliferative effect of the compound 49 authors did an MTT assay, cell cycle, and apoptosis assay with chloroquine. Furthermore, the effect of compound 49 on GSH-related enzymes was determined. In my recommendation, this manuscript requires major revisions before accepting it.

1.     Rewrite western blot data and explain results clearly

2.     Please explain why chloroquine was selected to exclude the role of autophagy.

3.     Why the effect of compound 49 on GSH-related enzymes was determined? Explain first and then discuss the results of compound 49.

4.     How GSH depletion is related to cisplatin, provide a few lines of explanation. What is concluded from the experiment?

5.     It is confusing to understand data with 2 different controls, chloroquine, and cisplatin. Results are not clearly presented.

6.     Rewrite the conclusion section. To provide a clear summary and what are the findings from the manuscript.

7.     It is advisable to present cell cycle and apoptosis data in graph format, to make it easy for readers to understand the data.

8.     If the author concludes compound 49 has an antiproliferative effect, please include western blot data for proteins involved in cancer cell proliferation.

9.     Include the IC50 value of the control in table 1.

Author Response

(The authors gave the same response as above.)

Reviewer 3 Report

The synthesis of novel bis-indole analogues with a phenyl linker obtained from indole phytoalexins and their in vitro antiproliferative activities have been studied by the authors.

1.       Title can be more precise and accurate.

2.       Is there a specific rationale for why this molecule was developed to combat lung cancer?

3.       Authors should discuss and explain why compound 49 was chosen for further research on the A549 lung cancer cell line. Is there a chemical, molecular, or synthetic reason for this? Because chemical 52a appears to be more potent than compound 49.

4.       Kindly provide the images of blots with a standard ladder along with molecular weight to compare the detected proteins.

5.       Why authors did not study BAX/BCL2/CASPASE to confirm apoptosis than is rather Annexin V/PI staining, this could have given more confirmatory results.

6.       Autophagy is a complex process; the findings showed just 5 protein markers and did not identify others. They should discuss alternative markers before concluding that the compound promotes autophagy.

7.       List all the limitations of your studies. 

Author Response

(The authors gave the same response as above.)

Round 2

Reviewer 2 Report

Thanks to the author for addressing all comments.